# Challenges in the Differential Classification of Individual Diagnoses from Co-Occurring Autism and ADHD Using Survey Data

Aditi Jaiswal
*Information and Computer Sciences Department,*
*University of Hawaii at Manoa*
Honolulu, USA
ajaiswal@hawaii.edu

Dennis P. Wall
*Departments of Pediatrics &*
*Biomedical Data Science,*
*Stanford University*
Stanford, USA
dpwall@stanford.edu

Peter Washington
*Information and Computer Sciences Department,*
*University of Hawaii at Manoa*
Honolulu, USA
pyw@hawaii.edu

*Abstract*—Autism and Attention-Deficit Hyperactivity Disorder (ADHD) are two of the most commonly observed neurodevelopmental conditions in childhood. Providing a specific computational assessment to distinguish between the two can prove difficult and time intensive. Given the high prevalence of their co-occurrence, there is a need for scalable and accessible methods for distinguishing the co-occurrence of autism and ADHD from individual diagnoses. The first step is to identify a core set of features that can serve as the basis for behavioral feature extraction. We trained machine learning models on data from the National Survey of Children's Health to identify behaviors to target as features in automated clinical decision support systems. A model trained on the binary task of distinguishing either developmental delay (autism or ADHD) vs. neither achieved sensitivity >92% and specificity >94%, while a model trained on the 4-way classification task of autism vs. ADHD vs. both vs. none demonstrated >65% sensitivity and >66% specificity. While the performance of the binary model was respectable, the relatively low performance in the differential classification of autism and ADHD highlights the challenges that persist in achieving specificity within clinical decision support tools for developmental delays. Nevertheless, this study demonstrates the potential of applying behavioral questionnaires not traditionally used for clinical purposes towards supporting digital screening assessments for pediatric developmental delays.

*Keywords—machine learning, autism, ADHD, survey data, differential diagnostics, feature selection*

## I. INTRODUCTION

Autism is often characterized by the presence of restrictive or repetitive behaviors, asynchronous emotions, sensory sensitivities, and atypical social communication patterns. By contrast, attention deficit hyperactivity disorder (ADHD) is defined by difficulties in attention and impulse control [1]. Although the two conditions differ in their diagnostic descriptions, it is frequently noted that a child with ADHD may exhibit characteristics of autism and vice versa. Prior research studies have investigated the prevalence rates of the co-occurrence of autism and ADHD in both children and adults, suggesting that 20-50% of individuals have a diagnosis of both conditions, and conversely, 30-80% of individuals diagnosed with autism have concurrent ADHD [2-4]. While the individual diagnosis itself is time intensive, the behavioral overlap between the two conditions further complicates differential classification.

Distinguishing between a diagnosis of autism only, ADHD only, and both remains challenging for both clinicians and computers. In recent years, machine learning (ML) has been used to identify salient reduced feature subsets for classifying autism with a minimum amount of information [5-

7], a more direct approach than other autism diagnostics research leveraging genotypic [8,9], phenotypic [10-13], and brain imaging data [14,15]. The results of such feature selection procedures have often been used as the basis for diagnostic crowdsourcing-enabled human-in-the-loop AI workflows [16,17], where the outputs of human annotations of the reduced behavioral feature set when observing a home video are fed as input into a classical ML model to perform a final diagnosis [18]. The generalization of human-in-the-loop ML to classify a broader range of pediatric developmental delays has yet to be studied. The first step to enable such workflows for multi-class diagnostics is to identify a salient list of behavioral features that can be analyzed and fed into a classical ML model.

Most, though not all, prior ML studies have focused on either a single binary diagnosis, enhancing the individual sensitivity of autism [19-22] and ADHD [23-26] classifiers or comparing the two conditions without considering co-occurring diagnoses. A few prior studies have examined the differential diagnoses between the two conditions [5,14,27,28]. These works demonstrate the potential of machine learning to optimize the features for simultaneous diagnoses that could help to build accessible screening platforms. However, these findings focus on gold standard clinical assessments, developed primarily for autism detection, limiting the availability of data related to ADHD.

Building upon this rich body of prior work, we aim to identify a subset of behavioral features from the publicly available National Survey of Children's Health (NSCH) data that could potentially be used for differential diagnosis of autism and ADHD, including the possibility of co-occurrence. Using population-based data instead of gold standard clinical assessments not only provides more data samples but also enables the analysis of a potentially more diverse and representative sample of the population. While some prior works have used similar data, they either focused on ADHD classification alone [25] or provided statistical analysis using socio-demographic features [8,29-31] rather than building ML models. The objective of our work, by contrast, is to identify the distinct subset of features that can effectively capture the complex behavioral patterns differentiating autism and ADHD co-occurrence between a diagnosis of only a single condition.

## II. METHODS

### A. Data Description

We used publicly available data from the National Survey of Children's Health (NSCH), a project overseen by the Health Resources and Services Administration Maternal and

Child Health Bureau. This survey targets children aged 0 to 17 years and offers comprehensive national and state-level insights into various facets of child health and emotional well-being. Responses to questions on various social determinants of the child are completed by the parents or guardians of the children who live in the same household.

We used data from the years 2016 (n=50,212), 2017 (n=30,530), 2018 (n=29,433), 2019 (n=42,777), 2020 (n=50,892), 2021 (n=21,599) and 2022 (n=54,103). For the target variables, we used the survey questions: "*Has a doctor or other health care provider EVER told you that the Selected Child (SC) has Autism or Autism Spectrum Disorder (ASD)?*" and "*Has a doctor or other health care provider EVER told you that SC has Attention Deficit Disorder or Attention-Deficit/Hyperactivity Disorder, that is, ADD or ADHD?*" If the answer to either question was yes, we encoded the output as "autism only" or "ADHD only", respectively. If the answers to both the questions were yes or no, they were encoded as "Both autism and ADHD" and "None", respectively. A similar strategy was used to categorize Learning disability, Depression, Anxiety, Behavioral problems, Developmental delay, Speech disorder, Tourette syndrome, and Intellectual disability.

*B. Data Cleaning*

The initial analysis of the data revealed that numerous behavioral features contained missing values. This could be attributed to respondents either deeming certain questions inapplicable or choosing not to disclose information, resulting in skipped responses. Records missing any target labels for diagnosis such as Tourette's syndrome, autism, ADHD, anxiety, depression, speech disorder, learning disability, and developmental delay were excluded to ensure the reliability of labels for ML model training. We additionally used sociodemographic variables to ensure balanced representation across different groups.

The filtered dataset resulted in 270,978 data points and 365 feature columns. As the study's focus was on identifying behavioral markers for autism and ADHD diagnosis, we discarded all the features not describing any observable behavior. Given the differing proportions of missing values across these features, we partitioned them into two distinct groups (Table I) and assessed their significance in classification tasks using ML algorithms.

TABLE I. DESCRIPTION OF THE FEATURE SETS USED FOR THE ML CLASSIFICATION TASKS

| Feature group 1 | Age, Sex, Difficulty concentrating, remembering or making decisions, Difficulty walking or climbing stairs, shows interest and curiosity in learning new things, Works to finish tasks started, stays calm and in control when challenged, argues too much, Difficulty making or keeping friends |
|---|---|
| Feature group 2 | Age, Sex, Shows interest and curiosity in learning new things, Difficulty making or keeping friends, Difficulty in coordination or moving around, Is affectionate and tender, Bounces back quickly when things do not go his or her way, Smiles and laughs a lot, Recognize beginning sound of a word, Recognize letters of alphabet, Gives good explanation of things he/she did, Can write his/her first name, Is easily distracted, Plays well with others, Shows concern when others are hurt or unhappy |

*C. Machine Learning Classification*

Three separate classification tasks were performed on each feature group: (1) a binary classification model for distinguishing individuals with any neurodevelopmental condition from those without any condition, (2) a binary classification model to distinguish individuals with either autism or ADHD from those without any condition, and (3) a multilabel classification model to distinguish autism only, ADHD only, both conditions, and none, generating a probabilistic prediction. For all three models, the dataset was divided into training, validation, and test sets using an 8:1:1 split. Logistic regression (LR) and decision tree (DT) models were used to assess the performance of each classification task independently.

The raw dataset is available on the official United States Census Bureau website via www.census.gov/programs-surveys/nsch/data/datasets.html. All of our machine learning code is available on GitHub via github.com/ucsfdigitalhealth/NSCH_feature_selection.

*D. Feature Selection*

We combined all features to assess their collective performance and identify which features contribute the most in the predictive tasks. Because some features contained numerous missing values, we applied imputation using a kNN Imputer, leveraging the Euclidean distance matrix to estimate missing values based on the closest data samples. Subsequently, we performed all three classification tasks on this combined feature set. We used the MLxtend sequential forward feature selector (SFS) implementation with a Random Forest (RF) classifier to identify the top 12 features yielding the highest classifier performance as measured by the Area Under the Receiver Operating Characteristic (ROCAUC) curve. Utilizing the reduced feature subset identified, we retrained the model on the original dataset to compare its performance when using all features. The overall workflow of the study is illustrated in Fig. 1.

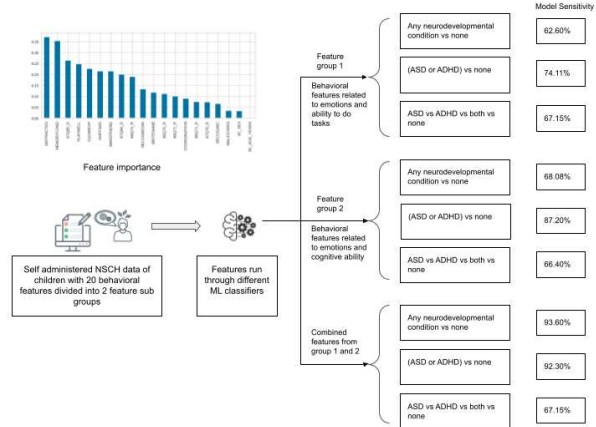

Figure 1. An overview of the steps taken for feature selection and machine learning classification using the NSCH data.

III. RESULTS

*A. Preliminary Data Analysis*

We conducted a preliminary data analysis on our filtered dataset to understand the sex, race, and age distribution among children with autism, ADHD, and both conditions.

The demographic breakdown of the dataset is summarized in Table II.

TABLE II.     DESCRIPTIVE STATSITICS OF THE SUBSET OF THE NSCH DATA WE EXAMINED

| Variable | Overall | Autism only | ADHD only | Autism + ADHD |
|---|---|---|---|---|
| Mean age (years) | 9 | 9.5 | 12.3 | 11.9 |
| Female | 130745 (48.3%) | 971 (22.5%) | 7840 (34.3%) | 773 (20.5%) |
| Male | 140233 (51.7%) | 3344 (77.5%) | 15023 (65.7%) | 2997 (79.5%) |
| White | 208993 (77.1%) | 3167 (73.4%) | 18400 (80.5%) | 2995 (79.4%) |
| Black or African American | 18175 (6.7%) | 370 (8.57%) | 1759 (7.7%) | 264 (7%) |
| American Indian or Alaska Native | 2424 (0.9%) | 43 (0.99%) | 212 (0.93%) | 33 (0.89%) |
| Asian | 15140 (5.6%) | 239 (5.53%) | 382 (1.6%) | 123 (3.3%) |
| Native Hawaiian and Other Pacific Islander | 1455 (0.54%) | 22 (0.51%) | 73 (0.32%) | 19 (0.5%) |
| Multiple races | 22232 (8.2%) | 430 (9.96%) | 1894 (8.3%) | 318 (8.43%) |
| Other | 2559 (0.96%) | 44 (1.04%) | 143 (0.6%) | 18 (0.477%) |

## B. Machine Learning Classification and Feature Selection

The classification model demonstrates decent performance in correctly distinguishing individuals with any neurodevelopmental condition from individuals with no condition as well as those with autism or ADHD versus neither. However, the performance diminishes for the 4-way classification task as the model tends to erroneously confuse autism with ADHD, and vice versa. Upon closer examination, we found that the probabilities associated with each prediction were often very similar, leading to confusion in the model's predictions (Fig 2).

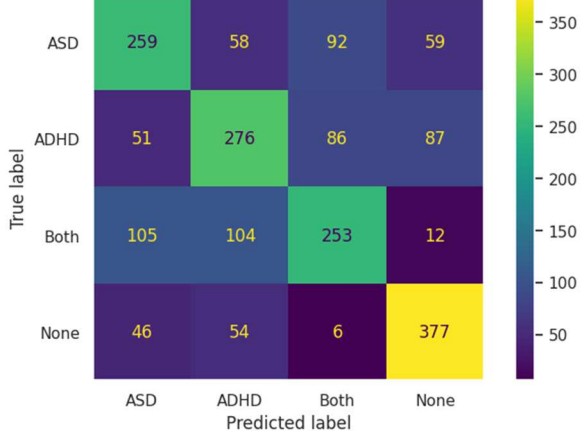

Figure 2. Confusion matrix for predictions specific to class Autism, ADHD, both and none.

Table III presents the evaluation metrics obtained on the test sets for each classification task using individual feature groups and the combined feature set using LR as the better performing classifier.

TABLE III.     RESULTS FOR THE 3 ML CLASSIFICATION TASKS ON EACH FEATURE GROUP

| Features used to train the model | Metric | Any neuro-developmental condition vs None | Autism or ADHD vs None | Multilabel autism + ADHD |
|---|---|---|---|---|
| Feature group 1 | Accuracy | 72.78% | 81.68% | 68% |
| | Sensitivity | 62.60% | 74.11% | 67.15% |
| | Specificity | 83.62% | 89.30% | 66.12% |
| | ROC-AUC score | 0.73 | 0.817 | Autism class: 0.83 ADHD class: 0.73 |
| Feature group 2 | Accuracy | 73.30% | 89.70% | 73.40% |
| | Sensitivity | 68.08% | 87.20% | 66.40% |
| | Specificity | 78.70% | 91.80% | 63.50% |
| | ROC-AUC score | 0.734 | 0.895 | Autism class: 0.82 ADHD class: 0.74 |
| Combined features | Accuracy | 93.80% | 93.50% | 68% |
| | Sensitivity | 93.60% | 92.30% | 67.15% |
| | Specificity | 94.04% | 94.70% | 66.12% |
| | ROC-AUC score | 0.94 | 0.93 | Autism class: 0.83 ADHD class: 0.73 |

Using forward feature selection on the combined feature set, the top 12 features identified were: age, sex, difficulty in making friends, difficulty in coordinating or moving around, affectionate and tender behavior, recognition of alphabets, easily distracted behavior, ability to play well with others, showing concern when others are hurt or happy, serious difficulty in concentrating or remembering things, tendency to work to finish tasks started, and arguing too much. Interestingly, these same features were consistently highlighted when training models using individual feature groups, further validating the model's performance on the combined feature set. In a series of separate studies [5,32] using a dataset containing responses from the Social Responsiveness Scale [33], researchers found certain features related to social interaction and attention regulation overlapping with the NSCH dataset, helping to partially validate the contribution of the behavioral features available in the NSCH survey.

Our analyses indicate that certain questions related to emotion and social skills contribute significantly to diagnostic uncertainty. Specifically, the feature 'ability to make friends' ranks as the second most important variable with an importance score of approximately 1.4, indicating its strong impact on the model's predictions. In contrast, items such as 'stays calm and in control when challenged' and 'argues too much' show moderate importance, with scores

around 0.6 and 0.5, respectively. These differences suggest that while emotional regulation factors contribute to the model, social interaction abilities play a more critical role in diagnostic accuracy. Using the partial dependence plots, we found that the model is more sensitive to changes in memory-related conditions when diagnosing ADHD than it is for ASD, making it more prone to diagnostic confusion and potentially leading to over- or under-classification in certain cases. However, despite the adverse impact on model sensitivity, this misclassification underscores the model's capacity to discern patterns indicative of neurodivergence broadly defined.

## IV. DISCUSSION AND CONCLUSION

This study emphasizes the importance of establishing behavioral markers that can distinguish between neurodevelopmental conditions with overlapping features by leveraging population-based questionnaire data. Using an optimal subset of behavioral features can aid in preliminary screenings of neurodevelopmental delays. Using the NSCH data, the multilabel classifier demonstrates moderate accuracy in distinguishing between autism and ADHD. It is important to note that although misclassifications may occur, the likelihood of the individuals falling into one of the two categories, if either category is classified, is very high. This aligns with prior work demonstrating that ML has the ability to identify some neurodivergent patterns in NSCH survey data that could aid in screening procedures [5,32].

There are several limitations of this study that must be noted. Importantly, the majority of the dataset is from White participants, with several groups disproportionately underrepresented. This can lead to potential model biases and thus emphasizes the need for high-quality, clinically annotated datasets with information from diverse populations. Additionally, the NSCH dataset is a survey of various aspects of children's physical, mental, and emotional well-being, with a focus on social and healthcare access factors. This contrasts with dedicated diagnostic tools such as the Autism Diagnostic Observation Schedule (ADOS) [34] and the Autism Diagnostic Interview-Revised (ADI-R) [35], designed to gauge autism spectrum severity, and the Conners Abbreviated Symptom Questionnaire [36], aimed at comprehensive ADHD assessment. While some observational behaviors from the NSCH dataset align with existing diagnostic tools, we acknowledge that the lack of external validation of our trained model and limited evaluation of feature selection are limitations of this study. We are actively working towards accessing more comprehensive datasets to further refine our model across different demographic groups and clinically validated features.

Additionally, the NSCH dataset contains several missing values. We tried to account for this by reducing the sample size and using model-based imputation techniques. The data loss that arises from fewer samples and discrepancies between the true and imputed values may negatively impact the model's predictive capacity. It is also important to acknowledge potential recall bias inherent in behavioral data obtained from parental questionnaires, and addressing these require datasets that use clinical questionnaires. However, such datasets with corresponding labels of both autism and ADHD are sparse. Finally, it is important to note that co-occurrence is not always diagnosed. A lack of a formal diagnosis does not necessarily mean that the child should not have a diagnosis. Because underdiagnosis of both autism and ADHD is a well-documented phenomenon [37-39], it is likely that there were children in the dataset who were reported to only have one diagnosis but should have had both, and some reportedly neurotypical children should have actually had at least one diagnosis.

In our future research, we will work on incorporating annotations of behavioral features that can distinguish autism, ADHD, and concurrent autism and ADHD into human-in-the-loop AI models [40]. The features identified in this study will be requested from human annotators, whose ratings will be fed as input into a ML model that will perform the final diagnosis. We ultimately aim to identify core behavioral features for early screening, thus helping to streamline clinical assessments and reduce time-consuming evaluation processes for autism and ADHD. Achieving clinical-grade performance in distinguishing the diagnosis of only autism or ADHD from having a diagnosis of both conditions poses a significant computational challenge. Solving this task is necessary given the high prevalence of concurrent autism and ADHD, and this work provides an initial pass at this problem using parent-reported behaviors.

## ACKNOWLEDGMENT

PW acknowledges funding from the National Institutes of Health (NIH) via the NIH Director's New Innovator Award (award DP2-EB035858). We used the generative artificial intelligence tool ChatGPT by OpenAI only to help with editing the grammar and phrasing of the text, and we further refined its output.

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
