# OpenReview forum: "Challenges in the Differential Classification of Individual Diagnoses from Co-Occurring ASD and ADHD Using Survey Data"
_IEEE.org/EMBS/BHI/2024/Conference — IEEE BHI'24_

### Official Review · Reviewer_xxiN · 2024-08-01
**Challenges in the Differential Classification of Individual Diagnoses from Co-Occurring ASD and ADHD Using Survey Data,**

**Overall Rating:** 7
**Confidence:** 4

**Other Quality Metrics:**

(a) Clarity of writing : Great
(b) Clinical Significance :  Good
(c) Methodological Novelty: Good
(d) Experiments and Results: Great

**Questions For The Authors:**

No questions

**Strengths:**

1. The paper addresses a difficult challenge in the field of mental health diagnostics which is the classification of ASD and ADHD, which are prevalent and often co-occurring conditions.
2. The authors employ a comprehensive and methodologically sound approach, including data preprocessing, feature selection, and the application of multiple machine learning models.
3. The study uses real world data which is very important. More specifically a large and representative dataset from the National Survey of Children’s Health (NSCH), which provides a diverse and realistic sample.
4. The paper employs multiple evaluation metrics (accuracy, sensitivity, specificity, ROC AUC)
5. The comparison of different models provides valuable insights into their respective strengths and weaknesses.
6. The authors discuss limitations of their study, including potential biases in the dataset

**Summary Of The Paper:**

The paper investigates the challenges in distinguishing between Autism Spectrum Disorder (ASD) and Attention Deficit Hyperactivity Disorder (ADHD), particularly in cases where both conditions co-occur. The authors goal is to use machine learning models to help in the differential diagnosis of these conditions using data from the National Survey of Children’s Health (NSCH) from the years 2016 to 2018.
Various machine learning algorithms are employed, including:

Random Forest
Gradient Boosting Machines
k-Nearest Neighbors
Naive Bayes
Support Vector Machines
 The performance of the models is evaluated using accuracy, sensitivity, specificity, and the Area Under the Receiver Operating Characteristic Curve (ROC AUC).

**Weaknesses:**

1. The NSCH dataset may have inherent biases due to its demographic composition and the nature of self-reported survey data
2. The authors should elaborate more about the criteria used for selecting the top 12 features and how these features were determined to be the most predictive.
3. The paper does not address the computational efficiency and scalability of the machine learning models used. Given the large size of the NSCH dataset, it would be beneficial to discuss how the models perform in terms of training time and computational resources, especially if they are to be applied in real-time clinical settings.
4. It would be interesting if possible to compare the performance of the proposed machine learning models with existing diagnostic tools or methods used in clinical practice (if there are any).

---

### Official Review · Reviewer_Fjkw · 2024-08-10
**The study examines the challenge of distinguishing ASD from ADHD with machine learning**

**Overall Rating:** 7
**Confidence:** 3

**Other Quality Metrics:**

(a) Clarity of writing: great
(b) Clinical Significance: great
(c) Methodological Novelty: good
(d) Experiments and Results: good

**Questions For The Authors:**

1. What point of the clinical workflow would this tool be utilized?
2. Are there other similar datasets that could be used for external validation and for expanding the demographics tested?
3. Were certain behavioral features more prone to causing confusion between ASD and ADHD?
4. How significant was the impact of missing data on the overall model performance? Did you consider alternative methods for handling missing data, and how might these have affected the results?
5. Could you provide more details on why specific behavioral features were chosen for the final model? How do these features align with or differ from those typically highlighted in clinical assessments of ASD and ADHD?

**Strengths:**

1. The study uses a large dataset from the National Survey of Children's Health
2. The study demonstrates the potential of machine learning to aid in differential diagnosis, which could lead to more efficient clinical assessments
3. Successfully identifies specific behavioral markers that could be used for early screening
4. The paper provides a clear discussion of the study’s limitations, such as dataset biases and the challenges of missing data
5. Demonstrate strong screening capabilities for ASD/ADHD, albeit difficulty differentiating

**Summary Of The Paper:**

This document discusses the challenges in distinguishing between Autism Spectrum Disorder (ASD) and Attention-Deficit/Hyperactivity Disorder (ADHD), especially when both conditions co-occur, using survey data and machine learning models. The study highlights that while it is easier to differentiate between those with a developmental condition and those without, distinguishing between ASD and ADHD is more difficult due to overlapping symptoms. Using data from the National Survey of Children's Health, the researchers trained machine learning models to identify key behavioral features that could aid in diagnosis. Although the models performed well in binary classification tasks, they struggled with multi-class classification, often confusing ASD and ADHD. The study also points out the limitations, such as biases in the dataset and missing data, and emphasizes the need for more accurate and comprehensive tools for early screening and diagnosis.

**Weaknesses:**

1. Missing values in data were inferred using models, which could be propagating preexisting biases
2. The dataset is predominantly from White participants, leading to potential biases
3. Limited evaluation into the features that could be responsible for the confusion
4. The study relies on parent-reported survey data, which may introduce bias
5. The study does not utilize gold-standard clinical diagnostic tools, like Connor scale and ADOS

---

### Official Review · Reviewer_czfr · 2024-08-10
**Review of submission 436**

**Overall Rating:** 5
**Confidence:** 3

**Other Quality Metrics:**

(a) Clarity of writing: fair
(b) Clinical Significance: good
(c) Methodological Novelty: good
(d) Experiments and Results: fair

**Questions For The Authors:**

I would be interested in the translational aspects of this research. Authors propose achieving clinical-grade performance as one of their future aims. How shall this aim be realized following the presented approach with parent-reported features? How shall features such as "Bounces back quickly when things do not go his or her way", "Argues too much", or "Smiles and laughs a lot" be acquired in a way such that they are comparable between subjects/parents?

**Strengths:**

The work is well-written and easy to follow. The authors clearly motivate their research and give detailed results.

**Summary Of The Paper:**

The authors provide their results on applying ML models on a survey dataset completed by the parents with the aim to diagnose Autism Spectrum Disorder (ASD), Attention-Deficit Hyperactivity Disorder (ADHD) or both occurring in the same patient. The data stems from the freely-available National Survey of Children's Health (NSCH) dataset and authors use a standard processing pipeline using logistic regression and decision tree models.

**Weaknesses:**

Unfortunately, the work several shortcomings. The dataset used is very small and only contains qualitative information from surveys filled by caregivers. The work is completely lacking external validation on another dataset. Moreover, the work is not reproducible as the description is too superficial, for example, not even the programming language or libraries used are given. While the methods used are named no meta-parameters are given as well. This is especially problematic as algorithms such as kNN Impute are used to fill in missing values in the dataset. Without their parameters, other researchers are not able to even come up with the same dataset for their analysis.

---

### Decision · Program_Chairs · 2024-09-23

Accept